# Moderate Weight Loss Modifies Leptin and Ghrelin Synthesis Rhythms but Not the Subjective Sensations of Appetite in Obesity Patients

**DOI:** 10.3390/nu12040916

**Published:** 2020-03-27

**Authors:** Juan José Hernández Morante, Inmaculada Díaz Soler, Joaquín S. Galindo Muñoz, Horacio Pérez Sánchez, Mª del Carmen Barberá Ortega, Carlos Manuel Martínez, Juana Mª Morillas Ruiz

**Affiliations:** 1Eating Disorders Research Unit, Catholic University of Murcia (UCAM), 30107 Murcia, Spain; inmadiaz93@hotmail.es (I.D.S.); mbobarbera@gmail.com (M.d.C.B.O.); 2“General Defence” Hospital, 50009 Zaragoza, Spain; joaquinsantiago.gm@gmail.com; 3Bioinformatics and High Performance Computing Research Group (BIO-HPC), Computer Engineering Department, Catholic University of Murcia (UCAM), 30107 Murcia, Spain; hperez@ucam.edu; 4IMIB-Experimental Pathology Service, Arrixaca Hospital, 30120 Murcia, Spain; cmmarti@um.es; 5Food Technology & Nutrition Dept., Catholic University of Murcia (UCAM), 30107 Murcia, Spain; jmmorillas@ucam.edu

**Keywords:** leptin, ghrelin, appetite sensation, hunger, satiety, circadian rhythm

## Abstract

Obesity is characterized by a resistance to appetite-regulating hormones, leading to a misalignment between the physiological signals and the perceived hunger/satiety signal. A disruption of the synthesis rhythm may explain this situation. The aim of this study was to evaluate the effect of dietary-induced weight loss on the daily rhythms of leptin and ghrelin and its influence on the daily variability of the appetite sensations of patients with obesity. Twenty subjects with obesity underwent a hypocaloric dietary intervention for 12 weeks. Plasma leptin and ghrelin were analyzed at baseline and at the end of the intervention and in 13 normal-weight controls. Appetite ratings were analyzed. Weight loss decreased leptin synthesis (*p*_auc_ < 0.001) but not the rhythm characteristics, except the mean variability value (*p*_mesor_ = 0.020). By contrast, the mean ghrelin level increased after weight loss. The rhythm characteristics were also modified until a rhythm similar to the normal-weight subjects was reached. The amount of variability of leptin and ghrelin was correlated with the effectiveness of the dietary intervention (*p* < 0.020 and *p* < 0.001, respectively). Losing weight partially restores the daily rhythms of leptin and modifies the ghrelin rhythms, but appetite sensations are barely modified, thus confirming that these hormones cannot exercise their physiological function properly.

## 1. Introduction

Despite the many advancements in the knowledge of the pathophysiological basis of obesity, this disease remains one of the main factors of disability and mortality in the world [1]. Although previous studies have suggested a relationship between obesity and the risk of chronic diseases (metabolic, cardiovascular, arteriopathies, mental diseases, or even cancer), the World Obesity Federation considers obesity as a chronic disease itself [2]. Nevertheless, regardless of how it is considered, the fact is that obesity may induce disability and early retirement with a significant increase in medical costs to the National Health System. Accordingly, the World Health Organization developed a plan to reduce the obesity prevalence in its Global Action Plan for the Prevention and Control of Non-Communicable Diseases 2013–2020 [3]. Therefore, controlling this pathology is mandatory nowadays.

Basically, the development of obesity is characterized by a dysregulation of the neurophysiological signals that control food intake, inducing an imbalance between energy expenditure and energy intake. One example of such neuroendocrine dysregulation is the development of resistance to leptin and ghrelin, two important endocrine regulators that develop their action mainly in the hypothalamus, where the activation of their respective receptors induces signaling cascades, which cause changes in food intake [4,5]. Leptin is produced in adipose tissue, is released to circulate in proportion with body fat mass [6], and comprises anorexigenic signals that mediate the long-term regulation of food intake, acting as a feedback mechanism that targets regulatory centers in the central nervous system (CNS) through the hypothalamus to inhibit food intake and regulate body weight and energy homeostasis [7,8]. Conversely, ghrelin is a hormone ubiquitously secreted by many tissues (stomach, gastrointestinal tract, ovary, adrenal cortex, and some regions of the CNS), and its secretion depends on the nutritional state by stimulated appetite, food intake, and weight gain [9]. As the targets of ghrelin in the CNS are the same as those of leptin (neuropeptide and agouti-related proteins in hypothalamic neurons), both hormones establish a complex regulatory pathway: ghrelin seems to reverse the inhibitory stimulus led by leptin to increase the appetite stimulus, whereas leptin antagonizes the increased food intake by ghrelin [10].

The complex molecular mechanisms of leptin resistance in obesity are not fully understood but seem to be selective, such as insulin resistance [11]. Several theories have been proposed to explain leptin resistance, such as the desensitization of the long form of the leptin receptor (LepRb) in individuals with obesity with chronically high levels of active free circulating leptin [12] or defective LepRb trafficking to the neuronal hypothalamic subpopulations that control energy homeostasis [13]. Similar to those of leptin, the mechanisms of ghrelin resistance in obesity are unclear. However, surprisingly, a positive dietary energy balance induces ghrelin resistance, and obesity is associated with a low release of circulating ghrelin, the levels of which do not decrease in response to a meal [14,15]. Although the molecular mechanisms that lead to leptin and ghrelin resistance are still under research, one interesting topic is the mechanisms associated with the disruption of the daily rhythms of factors driving the control of the hunger/satiety balance in humans [16]. Moreover, circadian rhythms seem to play a part in weight loss [17]. Thus, our previous report suggested that stress hormones (glucocorticoids) and their receptors control the daily cycles of adipose tissue metabolism by regulating gene transcription [18], a rhythm that increases in a high-fat diet [19]. Other reports have observed that ghrelin could restore the disruption of circadian rhythms in steatosis experimental models [20] and that circadian disruption could induce leptin resistance [21]. Although there are many reports establishing the disruption of circadian rhythms and the development of obesity, little is known about the restoration of such rhythms induced by weight loss. Moreover, it is not clear whether weight loss may modify the daily levels of leptin and ghrelin hormones and, in the event of changes, whether this corresponds to an alteration of the perception of hunger/satiety of subjects who have lost weight as a result of dietary treatment. Therefore, this study aimed to evaluate the effect of moderate weight loss as a consequence of a balanced hypocaloric diet on the daily synthesis rhythms of leptin and ghrelin and to determine if these rhythms also occur in the subjective sensations of hunger/satiety. As a secondary objective, we evaluated the relationship between the rhythmometric properties of appetite-related variables and the effectiveness of the weight loss dietary intervention.

## 2. Materials and Methods

### 2.1. Participants

Thirty-three participants (20 overweight and 13 normal-weight subjects) were selected to participate in the study through email and advertisements. Figure 1 shows the flow diagram of the study.

Subjects with smoking habits or those with alcohol consumption greater than 10 g/day were excluded. Individuals with a clinically significant illness that could affect the interpretation of data (cancer, human immunodeficiency virus (HIV), chronic conditions such as chronic obstructive pulmonary disease (COPD), cardiovascular disease (CVD), etc.) and cognitive-related disorders (epilepsy, Alzheimer’s disease, Parkinson’s disease, etc.) and those taking any medication known to affect body weight (thyroid hormones, corticosteroids, etc.) or appetite sensations (benzodiazepines, etc.) were excluded. All subjects were weight stable for at least three months. Thus, being under a dietary treatment for at least three months prior to participation in the study was also an exclusion criterion. 

The survey was carried out from January to December 2019, with previous written authorization from the ethics committee of the Catholic University of Murcia (#Code 101710). Patients were informed about the design of the study orally and in written form. An explanation of the research in the ethical sense was also given, informing the participants about the aim of the results obtained, the need for confidentiality and anonymity of the data, and respecting the Helsinki Declaration Agreement.

### 2.2. Study Design

This longitudinal experimental study was conducted to evaluate the possible changes in the daily circadian regulation of leptin, ghrelin, and the subjective appetite sensations of subjects undergoing a hypocaloric dietary treatment. 

At the beginning of the experiment (baseline), the subjects were interviewed at 08:00 h after an overnight fast to analyze the baseline appetite sensations and to determine the baseline clinical characteristics. An intravenous catheter was placed at this time. The participants were immediately invited to eat a fixed breakfast meal consisting of commercially available foods, with a relative contribution of energy from carbohydrates (71%), fat (20%), and protein (9%) (see Appendix A). Calories were estimated to deliver approximately 20% of the average daily energy expenditure. Four hours after the subjects finished their breakfast (13:00 h), they were provided with a fixed meal, with an average meal composition of 40% carbohydrates, 50% fat, and 10% protein. The meal was also composed of commercially available products, with a mean energy content of 785 Kcal (3218 Kjul). Five hours after the subjects had finished their lunch (19:00 h), a fixed dinner, with a similar composition to lunch, was delivered to the study participants. The total energy amount of the test meal was 2067 Kcal (8654 Kjul), with a macronutrient composition of 50% carbohydrates, 40% fat, and 10% protein. The subjects provided answers to appetite-related questions before and after every meal and every 1 h until the next meal. Blood pressure and heart rate were taken in the same way. Blood was drawn before and after every meal and every 2 h until the next meal (Figure 1). The catheter was removed when blood sampling was completed.

The subjects with a body mass index (BMI) equal or greater than 30 voluntarily followed the diet-induced weight loss program at the Nutrition Research Centre of the Catholic University of Murcia for 12 weeks. After this period, the patients were re-evaluated as described above to determine the possible changes in subjective appetite sensations, in the daily synthesis rhythms of leptin and ghrelin, and in other clinical characteristics. 

### 2.3. Hypocaloric Treatment

The dietary intervention was similar to that previously described [22,23]. Briefly, the patients were instructed to modify their usual diet with a balanced diet following a macronutrient distribution based on a Mediterranean diet consisting of carbohydrates (50%–60% of daily energy expenditure (DEE)), fat (30%–35% of DEE), and protein (15%–20% of DEE), following the FESNAD (Spanish Federation of Nutrition, Food and Dietetics) and SEEDO (Spanish Association for the Study of Obesity (SEEDO) guidelines [24]. The patients were monitored weekly to record their changes in body weight and body composition for 12 weeks. A dietitian (J.S.G.M.) designed the diets with the assistance of Dietowin 8.0 software (Bl-Biologica, Barcelona, Spain). This program includes the nutritional composition of more than 600 foods according to the tables of Spanish food composition. The program’s food database was modified to adapt some foods of frequent consumption and typical recipes of the Murcia region. The participants were suggested to perform 30 min of moderate aerobic exercise for at least five days a week (150 min/week) as recommended [25].

The hypocaloric diets in the group of subjects with obesity were structured in five meals in accordance with the following daily total energy intake distribution: breakfast 20%, mid-morning 10%, lunch 35%, mid-afternoon 10%, and dinner 25%. The maximum variability allowed in every meal was ±1%. The diets were designed according to the patients’ preferences, and unwanted meals were excluded from the menus. Total caloric intake was estimated through the subjects’ basal metabolic rate and physical activity level according to the FESNAD and SEEDO procedures [24]. The hypocaloric diets were designed based on a reduction of 1000 kcal/day to obtain a weight loss of 0.5–1 kg/week. Approximately, the daily energy estimation of the hypocaloric diets was 1600–2000 kcal/d for men and 1200–1500 kcal/d for women. The total energy expenditure was recalculated weekly, and the total energy expenditure of the hypocaloric diets was reformulated in the same base to accurately adjust to the energy requirements of the patients.

### 2.4. Evaluation of Subjective Appetite Sensations

The data on subjects’ subjective appetite sensations were obtained through a validated smartphone app [26]. Previous reports have shown that these devices are effective for monitoring and detecting changes in appetite sensations [27,28]. The new application (Dietavisa^®^) was presented in a BQ Aquarius 5 smartphone with three different screens for every question. The participants were instructed to read the question and move the cursor along a horizontal line. The cursor could be displayed by a single selection. Once the participants had confirmed that the cursor was in the ‘real’ subjective appetite sensation, they pressed the ‘continue’ button to confirm their response. The horizontal dimension of the new application scale had a length of 100 pixels (100 mm). 

The participants were presented with a series of questions accompanied by horizontal lines anchored at each end by the words ‘not at all’ (translated as ‘nada en absoluto’) and ‘extremely’ (translated as ‘extremadamente’). The order and wording of the sentences were as follows: ‘How hungry do you feel now? (¿Cuánta hambre sientes ahora?)’; ‘How full do you feel now? (¿Cómo te sientes de saciado ahora?)’; and ‘How strong is your desire to eat now? (¿Como de fuerte es tu deseo de comer ahora?)’. Each of the three questions was presented individually to the participants who were not able to progress to the next question until the current visual analogic scale (VAS) was correctly completed. 

### 2.5. Clinical Parameters

Blood samples were obtained before and after every meal and every 2 h until the next meal. The times of blood extraction were baseline (08:00 h), after breakfast (09:00 h), 11:00 h, before lunch (13:00 h), after lunch (14:00 h), 16:30 h, before dinner (19:00 h), and after dinner (20:00 h). This time series was selected because it was shown to be indicative of daily rhythms in previous studies conducted by our research group [26,29]. Leptin, ghrelin, and glucose were determined at these time points. The lipid parameters were determined only at baseline. The leptin and total ghrelin plasma levels were measured by fluorescence using the Human Dual-Range Leptin ELISA kit (EZHL-80SK) and the Human Ghrelin (total) (EZGRT-89K) ELISA kit (Millipore, Billerica, MA, USA) with a luminescence multi-plate reader (SpectraMax L, Molecular Devices, Sunnyvale, CA, USA). The sensitivity for the leptin kit was 0.125 ng/mL, and the intra-assay coefficient of variation (CV) was 2.6%–4.2%. For ghrelin, the sensitivity of the test was 30 pg/mL, and the intra-assay CV was 0.9%–1.3%. The glucose and lipid parameters were determined using commercial kits (Roche Diagnostics GmbH, Mannheim, Germany) following the manufacturer’s guidelines.

Systolic and diastolic blood pressure and heart ratio were determined using an OMRON M3-tensiometer (OMRON Healthcare Europe, Hoofddorp, The Netherlands). Anthropometric variables were assessed as stated by the criteria suggested by the SEEDO in 2007 [30]. Body weight, percentage of fat mass, and fat-free mass were measured by bioimpedance with a Tanita MC-780 MA® (Tanita Corporation of America, Inc., Arlington Heights, IL, USA). Height was measured with a Tanita rod (model: Harpender), and BMI was derived from these data. The distribution of body fat was analyzed using the measurement of waist circumference. Each measurement was performed three times in a non-consecutive manner by the same investigator.

### 2.6. Statistical Analysis

Population size was calculated using GPower software ( release 3.0, Dusseldorf, Germany) [31]. A priori power analysis for the F test was performed to control for both type 1 and type 2 probability errors. The sample size was estimated according to the variance observed in previous works [27,29]. A minimum sample size of 12 subjects was estimated. Given that previous works described a 25% drop out, the composition of the study group was *n* = 20 for the subjects with obesity (considering a 20%–30% dropout rate) and *n* = 13 for the control group. Our sample size yielded a greater than 80% power, which enabled true within-group differences to be detected with an effect size of partial η^2^ ≥ 0.5.

A basic descriptive analysis was performed. Student’s *t*-test was used to evaluate the baseline differences between the completers of the obesity group (*n* = 18) and those of the control group (*n* = 13). The comparison of leptin, ghrelin, and subjective appetite sensation levels with time between the two groups was analyzed using the repeated measures ANOVA test and the post-hoc test of Tukey’s correction. A mixed factorial design using time (pre- and post-values) as the within-subject factor and group (normal-weight or obesity) as the between-subject factor was conducted to analyze possible interactions (time × group) in the studied variables.

Circadian rhythms were examined by applying the multiple cosinor regression with the assistance of the Chronomics Analysis Toolkit (CATkit) [32], which quantitatively defines the rhythm in specific individuals, populations, or groups by grouping the variables (leptin, ghrelin, clinical parameters, and feelings of appetite) as a function of time. Its graphic representation, a simple harmonic equation, is expressed as follows: Y(t) = M + A cos (2πt/t + ϕ) + e, where Y is the variable of interest; t is the time period; M is the regression constant or Mesor, which is equivalent to the average value of the function; A is the amplitude (half the extent of predictable variation within a cycle); ϕ is the acrophase (a measure of the time of the overall high values recurring in each cycle); t is the period; and e is the error term. 

Statistical analyses were performed using GraphPad 7.0 software (GraphPad Software, San Diego, CA, USA) and R Software (R Foundation for Statistical Computing, Vienna, Austria). The level of significance for all statistical tests and hypotheses was set to *p* < 0.05.

## 3. Results

### 3.1. Clinical Characteristics of the Subjects Studied

The baseline characteristics of the control and obesity groups are shown in Table 1. As expected by the study design, all anthropometric variables were significantly higher in the obesity group. Age, sex distribution, and biochemical parameters were similar in both groups, and no statistical differences were observed. Only high-density lipoprotein-associated cholesterol (cHDL) showed a slight trend towards better values in the control subjects, but as mentioned, the differences did not reach statistical significance. 

After 12 weeks, the subjects lost a mean weight of −8.6 ± 3.2 kg, which represented a general percentage of total weight loss of 10%. BMI, body fat, and waist circumference decreased by −3.10 ± 1.02 kg/m^2^, −4.67 ± 1.89%, and −9.9 ± 2.8 cm, respectively. The weight loss also improved several biochemical and clinical parameters, specifically those with reference to triglycerides and diastolic blood pressure. The detailed changes and statistical significance values are shown in Table 2. 

### 3.2. Effect of Weight Loss on Appetite-Regulating Hormones

The daily synthesis rhythms of leptin and ghrelin were analyzed, and the data obtained are presented in Figure 2. The glucose levels were measured as internal controls. The main differences were observed in the daily synthesis of leptin between the control subjects and the patients with obesity before treatment (*p*_auc_ < 0.001). Interestingly, after 12 weeks of hypocaloric dietary treatment, the leptin synthesis rhythm was reduced to 36% (area under the curve (AUC) decreased from baseline) until the values of the control subjects and the treated subjects were matched, indicating an effect of weight loss on the daily synthesis of leptin. 

In contrast to leptin, the baseline daily synthesis rhythm of ghrelin was lower in the obese subjects (Figure 2c), and weight loss induced an increase in ghrelin synthesis post-treatment (18% from the baseline AUC). However, this increase did not reach statistical significance (*p*_auc_ = 0.241). Otherwise, the glucose rhythm was similar in all subjects, and the effect of weight loss was uncertain. 

### 3.3. Effect of Weight Loss on Subjective Appetite Sensations

Figure 3 shows the daily rhythm of subjective perception of hunger, satiety, and desire to eat before and after the hypocaloric treatment and in the control subjects. The normal-weight control subjects had a higher hunger sensation than those with obesity both before (*p*_auc_ = 0.003) and after the hypocaloric treatment *(p*_auc_ = 0.002). The results indicate that weight loss did not modify the daily hunger perceptions (Figure 3a). Moreover, satiety scores were similar in all the groups studied. In sum, the control subjects showed higher satiety scores, especially in the last hours of the day, but no inter-group statistical differences were found at any time (Figure 3b). The results on the desire to eat sensation followed a similar trend to the hunger variable (Figure 3c). 

### 3.4. Rhythmic Analysis of Leptin and Ghrelin and Their Relation to Appetite Scores

To characterize the synthesis rhythm of the studied variable, a cosinor analysis was performed to determine the rhythmometric characteristics of these appetite-related variables (Figure 4). Glucose rhythm was dependent on mealtime (Figure 4a). The daily rhythm of leptin was characterized by a similar synthesis peak at 08:00 h and 18:00 h, and the rhythm valley was at 12:00 h. From this point, the plasma leptin levels progressively increased to the last determination (20:00 h). At baseline, the leptin mesor was significantly higher in the obesity group subjects than in the control subjects. After weight loss, the leptin rhythm was practically identical to the control group (Figure 4b). In contrast to leptin, the baseline ghrelin rhythm was lower in the obesity group than in the control group. Weight loss induced an increase in the mean ghrelin level (mesor) as a significant modification of the daily rhythm of ghrelin (Figure 4c). Specifically, the peak of ghrelin synthesis was observed at 10:00 h at baseline in the subjects undergoing the dietary treatment. However, after the intervention, this point corresponded to the synthesis rhythm valley, which ultimately implies that weight loss modified the synthesis rhythms of ghrelin to cause variability patterns similar to those of the control subjects. 

The synthesis rhythms of leptin and ghrelin were plotted against the hunger/satiety scores to evaluate the relation among these variables (Figure 5). The results indicated a high morning amplitude for leptin secretion and hunger feelings between 12:00 h and 14:00 h (around lunch time, Figure 5a), especially in the control subjects. Interestingly, this amplitude was reduced in the obese subjects but was restored after weight loss. Regarding the subjects’ satiety perceptions, the results showed a parallel rhythm with leptin synthesis, as indicated in the decrease in both variables in the first hours of the day (Figure 5b). The relationship between ghrelin and appetite ratings was complex. For example, the control and post-treatment subjects showed an increase in ghrelin along with an increase in hunger (Figure 5c). Conversely, the rhythm acrophase of ghrelin in the obese subjects before treatment corresponded to the lowest sensation of hunger, indicating a misalignment between the physiological ghrelin action and the patients’ feelings of hunger. A similar trend was observed in satiety feelings (Figure 5d). However, the low variability of the ghrelin rhythm before treatment did not show a clear association between ghrelin and the feeling of hunger or satiety.

As one of our aims was to evaluate the relation between the daily variations of appetite-related variables and the effectiveness of the hypocaloric intervention, a correlation analysis was conducted between the anthropometrical variables and the variables describing the circadian rhythm properties. The results showed that the percentage of rhythmicity (PR%) is the most associated chronological variable in the anthropometrical indicators of weight loss (Figure 6). The PR% of leptin was statistically significantly correlated with changes in body weight, BMI, and waist circumference, whereas the PR% of ghrelin was related to a decrease in body fat (Figure 6). The detailed correlation coefficients of the statistically significant associations are shown in Appendix A. 

## 4. Discussion

Given that this study was conducted to evaluate the effect of moderate weight loss on the rhythmic properties of the daily synthesis of leptin and ghrelin and whether these rhythmic properties were related to the indicators of weight loss treatment effectiveness, the first step to answer these objectives was to identify the existence of such synthesis rhythms. Thus, a daily synthesis rhythm was observed for both leptin and ghrelin hormones, a situation that had been previously defined [33,34]. The daily levels of leptin were significantly higher in the obesity group, as previously described [35]. As leptin is an adipose-derived hormone, assuming that a reduction in body fat may reduce leptin synthesis is logical [36,37]. 

Although weight loss has been shown to reduce leptin levels [38], the effect of weight loss on the characteristics of the synthesis rhythms of leptin has not been well studied. The cosinor analysis performed in this study showed that the rhythmic properties were similar in all groups, as similar amplitude, acrophase, and percentage of rhythmicity were observed in all groups. However, the mesor of the rhythm significantly decreased, indicating that weight loss only modified the average leptin synthesis level but did not affect the rhythmic properties of leptin synthesis. 

Several authors have shown that bariatric surgery decreased the overall 24 h leptin levels in morbidly obese women [39,40]. Other studies, such as Mantele et al., carried out in populations with a more moderate degree of obesity, showed no differences in amplitude, cycle mean, or timing of leptin rhythms between lean and obese subjects [41], similar to the present work. Interestingly, Sofer et al. examined the influence of food intake timing on leptin reduction [42] and found that the subjects eating carbohydrates only at dinner were characterized by an anomalous decrease in leptin in the evening [42]. As the diet of the patients was controlled in the present work, with the same nutrients ingested at the beginning and after the intervention, we assumed that the observed changes in leptin rhythm were mainly due to weight loss. 

In spite of the abrupt decrease in daily leptin levels, hunger/satiety perceptions were not modified after the dietary intervention, confirming the leptin resistance of obese patients [43]. Potential mechanisms, such as a deficient movement of leptin through the blood–brain barrier, a defective leptin receptor translocation, and inflammation and endoplasmic reticulum stress, have been proposed [13]. Although it modified the leptin synthesis rate, the weight loss produced during the 12 weeks of intervention was not able to reverse these mechanisms.

Daily ghrelin levels were also modified after weight loss, but unlike the leptin hormone, an increase in daily ghrelin values was observed, although these differences did not reach a statistically significant level. Previous reports have shown conflicting results in this regard, indicating the lack of effect of diet-induced weight loss on the plasma ghrelin levels [44,45]. Pamuk et al. observed a slight increase in ghrelin levels after an orlistat-induced weight loss [46]. However, as these previous works only evaluated one specific ghrelin measurement, comparisons should be made with caution. The pioneer work of Cummings et al. [47], which evaluated daily ghrelin synthesis, described a significant reduction in ghrelin levels after gastric bypass surgery but an increase in ghrelin synthesis after diet-induced weight loss, consistent with the present work. The amount of weight loss may determine certain statistically significant changes in ghrelin levels, and greater weight loss is necessary to observe a statistically significant effect. 

Unlike those of leptin, the rhythmic properties of ghrelin synthesis were clearly modified, as the synthesis peak was observed in the afternoon (near 16:00 h) in both the control and obesity groups, as previously described [42], but only after weight loss. By contrast, at baseline, this time point corresponded to the nadir of the synthesis rhythm. Ghrelin is synthesized to initiate meals, as a pre-prandial rise in plasma ghrelin levels is observed in studies on a fixed meal schedule [48,49]. However, a synthesis rhythm was barely detected in obese subjects. Several theories may be involved in this observation, but an interesting theory comes from Briggs et al. [50]. These authors proposed hypothalamic inflammation as a potential mechanism of ghrelin resistance. Therefore, as obesity is considered a mild inflammation disorder [51], and weight loss reduces inflammation in obese subjects [52], the weight reduction observed in the present work could have reduced local inflammation and partially restored ghrelin resistance. Whatever the mechanism involved, and as previously suggested by Giammanco et al., these hormone alterations are secondary to obesity, so they may be reversible following body weight loss [53].

Despite the strong evidence asserting the biological adaptations of leptin and ghrelin after weight loss, which increase the propensity for weight regain [54], their changes, taken alone, are not sufficient modulators of the homeostatic regulation of appetite [38]. To understand this result, the hormones leptin and ghrelin should be considered as only one facet of a highly complex, systemic response that involves the CNS, gut, adipose tissue, and other organs [55,56]. The hedonic regulation of appetite, that is, the processes involved in the effect of reward, pleasure, and palatability on eating, may prevail over the physiological factors in obese subjects [57], and could have determined our observations. 

Since we first described the implication of the clock genes of human adipose tissue on obesity-related metabolic disturbances [58], the disruption of normal biological rhythms has attracted great attention in explaining obesity development [59]. Our previous report also demonstrated the relevance of adapting a hypocaloric treatment to the biological rhythms of patients [23]. Therefore, it is tempting to speculate that, with respect to weight loss, the effect of leptin and ghrelin rhythms is more important than that of the basal values of these hormones. This hypothesis is reinforced by the data obtained in the present work, in which the subjects with a greater daily variability of leptin and ghrelin were found to lose more weight and body fat. As we have not found previous studies to support these observations, future research should confirm the relation between the rhythm characteristics of leptin and ghrelin synthesis and patients’ ability to lose weight. 

This study has several limitations. Although baseline characteristics were similar in both groups, there was a higher number of women, a situation that may have influenced our observations. Only the diurnal rhythms of leptin and ghrelin were evaluated, and the effect of weight loss on the nocturnal rhythms of these hormones could not be ruled out. As our results coincide with those of previous studies on this nocturnal variability [34,47], we can assume a similar trend in our patients. Despite being key regulators of appetite, the modification of the rhythm characteristics of leptin and ghrelin hardly exerted any effect on the subjective sensations of appetite. This is not an unexpected scenario, as appetite is regulated by both biological and non-biological inputs that can influence these sensations. In fact, factors related to meal quality, such as expectation or reward and pleasure for food, may serve as determinants of satiety regulation more than biological factors [56,57]. On the other hand, the lack of significant effects on appetite sensation could simply be due to the limitation in this parameter, because it relies on subjective determination. In this line, the inclusion of a positive control may help to confirm our observations in future studies. 

## 5. Conclusions

In conclusion, losing weight ‘normalizes’ the daily rhythms of leptin and modifies ghrelin rhythms, but hunger/satiety sensations are barely modified. This result confirms that these patients suffer from appetite-regulating hormonal resistance, which causes physiological functions to not respond adequately to regulate appetite. Moreover, a relationship exists between the characteristics of these rhythms and the effectiveness of the dietary intervention, which can be useful for predicting a person’s ability to lose weight. Since this is an observational study, it would be interesting to confirm in further studies, with a higher number of participants, whether there is a causal relationship between resetting hormone rhythms and weight loss, and especially, if this relation is also preserved at long-term.

## Figures and Tables

**Figure 1 nutrients-12-00916-f001:**
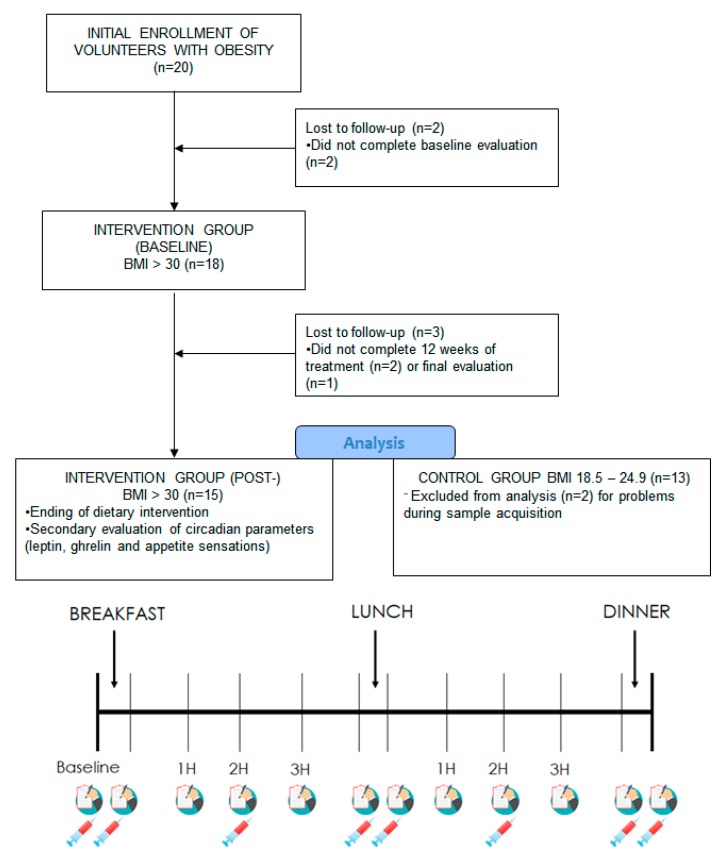
Flow diagram and study design. In the lower figure, the baseline indicates the first determination before having breakfast. 1H, 2H, and 3H indicate the time (in hours) after the end of the meal by part of the participants. BMI: body mass index.

**Figure 2 nutrients-12-00916-f002:**
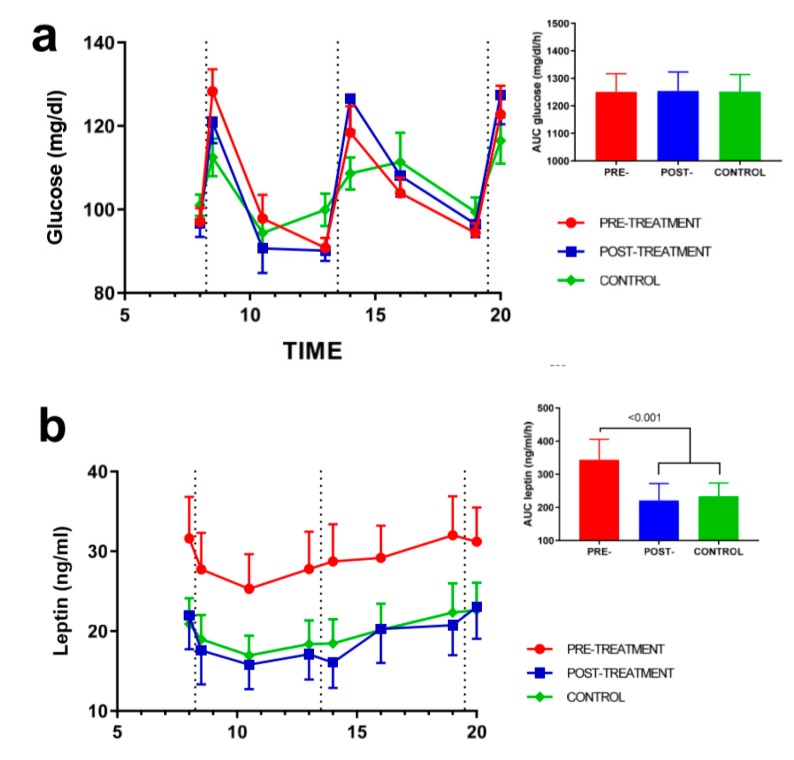
Diurnal synthesis rhythms of (**a**) glucose, (**b**) leptin, and (**c**) ghrelin in the control (green line) and obese subjects at baseline (red line) and after 12 weeks of hypocaloric dietary treatment (blue line). The vertical dotted lines indicate the time of breakfast, lunch, and dinner. Mean ± standard error The differences between the groups were analyzed by a two-way (time × group) repeated-measures analysis of variance (ANOVA).

**Figure 3 nutrients-12-00916-f003:**
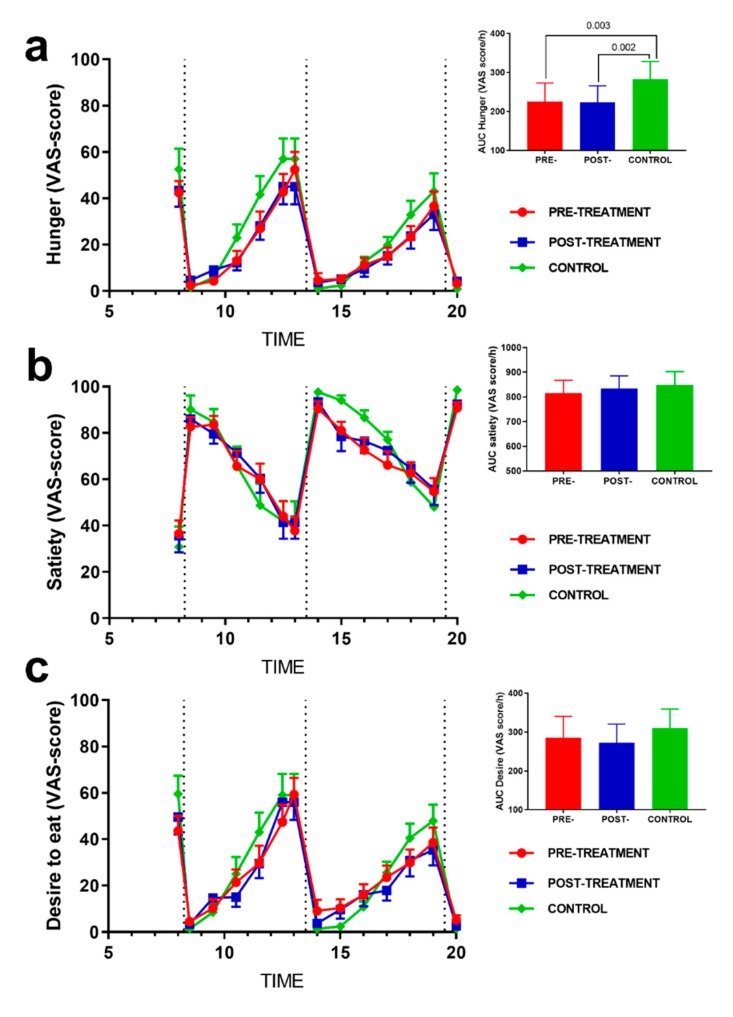
Comparison of (**a**) subjective hunger, (**b**) satiety, and (**c**) desire to eat ratings in the subjects with obesity pre- and post-intervention and in the control subjects. The vertical dotted lines indicate the time of breakfast, lunch, and dinner. Mean ± SEM. The differences between the groups were analyzed by a two-way (time × group) repeated-measures ANOVA.

**Figure 4 nutrients-12-00916-f004:**
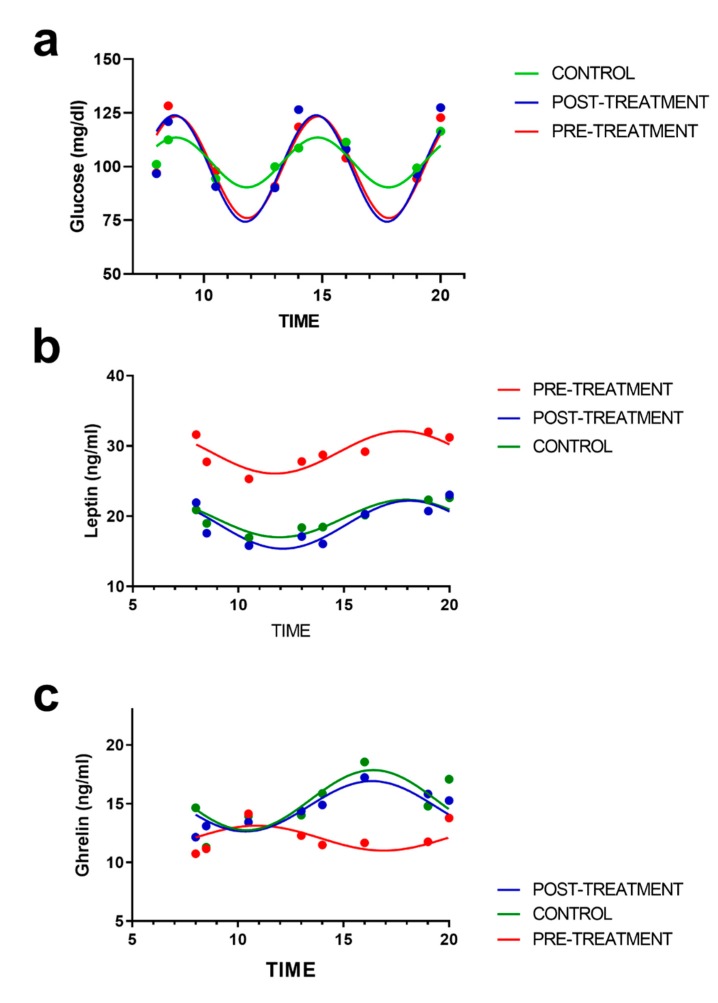
Circadian variation in the daily synthesis levels over time for (**a**) glucose, (**b**) leptin, and (**c**) ghrelin in the control (*n* = 13) and obese subjects at baseline (*n* = 18) and after 12 weeks of treatment (*n* = 15). Dots indicate the mean value. The curve represented for each group corresponds to the best-fitted model obtained by population multiple-component analysis.

**Figure 5 nutrients-12-00916-f005:**
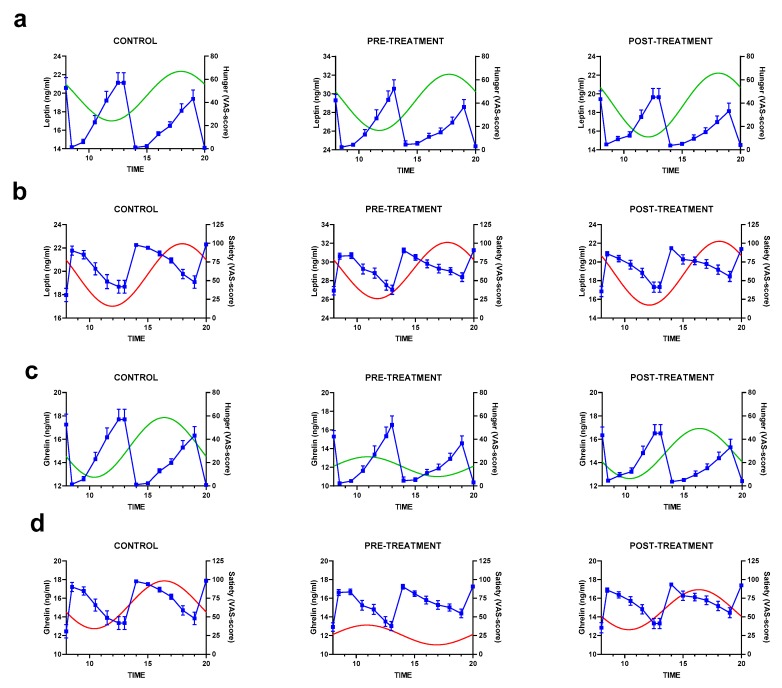
Overlaid cosinor fitted curve (left *y*-axis) and appetite sensations (right *x*-axis) for (**a**) leptin and hunger, (**b**) leptin and satiety, (**c**) ghrelin and hunger, and (**d**) and ghrelin and satiety. Hunger and satiety scores represent the mean ± SEM. Green and red lines represent leptin and ghrelin cosinor fitted curve of every group (control, pre- and post-treatment).

**Figure 6 nutrients-12-00916-f006:**
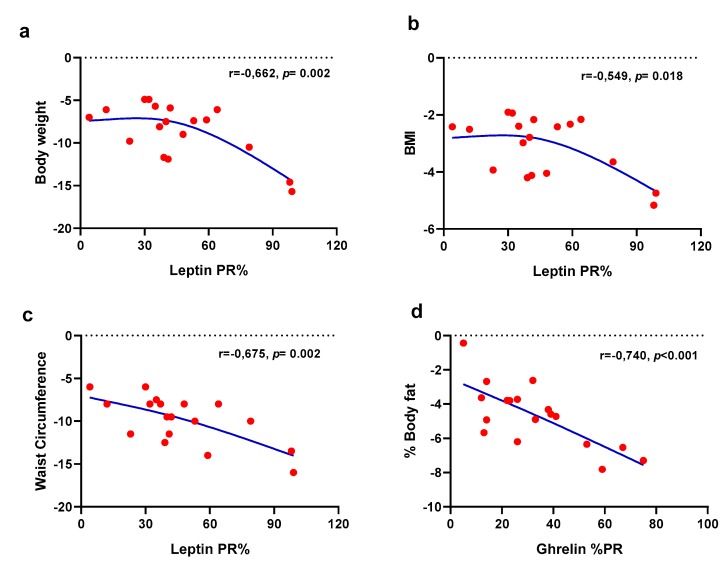
Association plots between the percentage of rhythmicity (PR%) of leptin and body weight (**a**), BMI (**b**), waist circumference (**c**) and the PR% of ghrelin and the body fat percentage (**d**) as indicators of the effectiveness of the weight loss dietary intervention. Pearson’s correlation analysis was used to obtain the correlation coefficients (r) and their associated significance level (*p*). Lines were smoothed using a smoothing spline with degree of freedom (*df*) = 3. The detailed description of the statistically significant correlation coefficients is presented in Appendix A.

**Table 1 nutrients-12-00916-t001:** Subjects’ characteristics at baseline.

	Control (*n* = 13)	Intervention (*n* = 18)	*p* (*t*,*χ*^2^)
Age (years)	44 ± 7	47 ± 9	0.222
Sex (*n* of women)	9	13	0.583
Weight (kg)	63.8 ± 8.1	85.4 ± 13.2	<0.001
BMI (kg/m^2^)	22.46 ± 2.11	31.14 ± 3.50	<0.001
Waist (cm)	83.0 ± 5.7115	102.6 ± 10.6	<0.001
Body fat (%)	26.08 ± 5.44	37.09 ± 5.93	<0.001
SBP (mmHg)	120 ± 13	126 ± 16	0.309
DBP (mmHg)	71 ± 9	75 ± 8	0.237
Fasting glucose (mg/dL)	101 ± 9	97 ± 13	0.382
Triglycerides (mg/dL)	103.2 ± 33.5307	114.1 ± 27.6	0.413
cHDL (mg/dL)	61.6 ± 23.7	49.4 ± 10.8	0.146

All values are the mean ± SD. Statistical significance was determined by the independent *t*-test, except sex distribution, which was calculated by the Chi-squared test. BMI: body mass index; SBP: systolic blood pressure; DBP: diastolic blood pressure; cHDL: high-density lipoprotein-associated cholesterol.

**Table 2 nutrients-12-00916-t002:** Change in clinical characteristics at baseline and after 12 weeks of intervention.

	Baseline (*n* = 18)	Post-Intervention (*n* = 15)	Change	*p*
Weight (kg)	85.4 ± 13.2	76.9 ± 11.0	−8.5	<0.001
BMI (kg/m^2^)	31.14 ± 3.50	28.04 ± 2.89	−3.10	<0.001
Waist (cm)	102.6 ± 10.6	92.7 ± 9.9	−9.9	<0.001
Body fat (%)	37.09 ± 5.93	32.42 ± 5.83	−4.67	<0.001
SBP (mmHg)	126 ± 16	120 ± 12	−6	0.146
DBP (mmHg)	75 ± 8	70 ± 8	−5	0.008
Fasting glucose(mg/dL)	97.1 ± 14.7	96.7 ± 12.8	−0.4	0.886
Triglycerides (mg/dL)	114.1 ± 27.6	90.0 ± 21.8	−24.1	0.034
cHDL (mg/dL)	49.4 ± 10.8	60.8 ± 16.6	+11.4	0.108

All values are mean ± SD. Statistical significance was determined by paired *t*-test.

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
