# Peer review of "Moderate Weight Loss Modifies Leptin and Ghrelin Synthesis Rhythms but Not the Subjective Sensations of Appetite in Obesity Patients"

_nutrients, 2020, doi:10.3390/nu12040916_

Round 1
Reviewer 1 Report
The work is well structured and only minor revisions have to be made. It would be useful to insert the following bibliographical note: Giammanco M., et al. Nutrition, obesity and hormones. Journal of Biological Research 2018;91:108-18.
Author Response
"Please see the attachment."

Reviewer 2 Report
The paper by Morante et al on leptin and ghrelin daily rhythms describes an extremely interesting observation, namely that weight loss of about 10% of weight results in a reset of the metabolism, in line with similar findings of insulin regulation (shown in rats and suspected in humans previously).
I did not completely agree with the conclusions on the appetite sensations. The lack of significant effects on appetite sensation could simply be due to the limitation in this parameter, because it relies on a highly subjective questionnaire. If the authors want to include this negative result, they need to demonstrate that the methodology used is actually able to produce a positive result, i.e. a positive control needs to be devised.
Some minor comments:
In figure 1, the legend needs to be expanded to explain what does 1H, 2H and 3H refer to and what is the baseline.
For the Ethics approval – is there a reference number? Usually there is a specific protocol that can be referred to.
I think that the authors should point out and discuss the possible implications that (1) the study involved approximately 2/3 women, and (2) that the number of participants was relatively low and thus the overall validity awaits validation with larger numbers of participants.
Since this is an observational study, it would be interesting to discuss the design of a future clinical trial that tests the inverse of the derived hypothesis, namely that there may be a causal relationship of reset hormone rhythms results in weight loss. It would also be interesting to study how long the reset takes. Since in the current study only before and after 12 week diet values are presented, it would be interesting to see how long it takes for this effect to become visible (e.g. measure every week). This could then be used as a motivating factor for people dieting to explain to them that the beneficial effects require x number of days/weeks to kick in.
Another interesting idea that the authors only touched upon and that could be further elaborated is the idea of personal differences. Are the larger variations in hormone concentrations a cause or effect of the weight loss?
Author Response
"Please see the attachment."

Reviewer 3 Report
The study aims to evaluate the effect of weight loss by defined and balanced hypocaloric diet on the daily synthesis of leptin and ghrelin. Levels of leptin and ghrelin have been reported in obese patients. Their circadian rhythms also have been studied in obese humans elsewhere including a few cited references and the followings (Gomez Abellan P et al. Nutr Hosp. 2011, 26(6):1394-1401, Yildiz BO et al. PNAS. 2004, 101(28):10434-10439). The current study added subjective parameters such as hunger, satiety, and desire to eat into the diurnal rhythms and their changes after 12 week balanced hypocaloric diet regimen, where a novelty exists. The study conclusion is that moderate body weight loss changes the diurnal rhythms of leptin and ghrelin to normal level but not the subjective sensations of appetite in obese human subjects. It is noteworthy that the underlying mechanisms for the resistance of anorexigenic effect of leptin have been recently reported in rodents (Mazor R et al. Sci Transl Med. 2018, 10(455):eaah6324). The current manuscript described the methodology in detail and discussed their observations and limitations clearly. Here are some minor issues the authors should address.
In 2.1, the number of participant women is 22 not 33 according to Table 1.
Table S1 should be a main table.
Difference between hunger scores is significant only when presented AUC? How about each time point? In addition, errors of those subjective appetite scores are supposed to be bigger than normal experimental errors? Thus how statistical significance is defined in a subjective measurement?
Label lines on the graphs accordingly in Fig. 5.
Author Response
"Please see the attachment."

Round 2
Reviewer 2 Report
I am fine with the responses to my comments and the revisions made.